# Combinatorics and Statistical Mechanics of Integer Partitions

**DOI:** 10.3390/e25020385

**Published:** 2023-02-20

**Authors:** Themis Matsoukas

**Affiliations:** Department of Chemical Engineering, Pennsylvania State University, State College, PA 16801, USA; txm11@psu.edu; Tel.: +1-814-863-2002

**Keywords:** integer partitions, partition function, statistical mechanics

## Abstract

We study the set of integer partitions as a probability space that generates distributions and, in the asymptotic limit, obeys thermodynamics. We view ordered integer partition as a configuration of cluster masses and associate them with the distribution of masses it contains. We organized the set of ordered partitions into a table that forms a microcanonical ensemble and whose columns form a set of canonical ensembles. We define a functional of the distribution (selection functional) that establishes a probability measure on the distributions of the ensemble, study the combinatorial properties of this space, define its partition functions, and show that, in the asymptotic limit, this space obeys thermodynamics. We construct a stochastic process that we call exchange reaction and used it to sample the mean distribution by Mote Carlo simulation. We demonstrated that, with appropriate choice of the selection functional, we can obtain any distribution as the equilibrium distribution of the ensemble.

## 1. Introduction

The central element of statistical mechanics is the ensemble and its partition function. The microcanonical ensemble contains all microstates with fixed macroscopic energy, volume, and number of particles. The canonical ensemble is a subset of the microcanonical (“system”), and its complement forms another canonical ensemble (the “bath”). The microcanonical and the canonical partition function are Legendre transformations of each other, and their derivatives generate all thermodynamic properties. This summarizes the thermodynamic formalism of Gibbs [1], a recipe that has proven immensely successful in physics, chemistry, and biology. At the heart of Gibbs’s method is the enumeration of microstates, a primitive stochastic variable whose enumeration (multiplicity) is a key element of the method. In the microcanonical ensemble, all microstates are represented in equal numbers; in the canonical ensemble, they are represented in proportion to the Boltzmann factor $e^{-E/k_{B}T}$, as the same microstate can be paired with several microstates in the bath. These multiplicities fix the probability of the microstate. We may generalize this method and apply it to other types of ensembles of “configurations” that are not physical microstates. The set of integer partitions provides a concrete example of an ensemble whose elements can be enumerated exactly. A partition of integer *M* into *N* parts is a list of *N* positive integers whose sum is *M*. This list can be viewed as *M* monomers combined into *N* clusters (polymers). The number of monomers, dimers, etc., present in the partition represents a possible distribution of clusters. The set of partitions forms an ensemble defined by the macroscopic variables *M* and *N*, which, like energy, volume, and the number of particles in statistical mechanics, act as constraints on the microstates that can be accessed by the macroscopic system. Integer partitions appear naturally in population balance problems. In discrete fragmentation [2,3,4], fragments are partitions of the mass that generates them. In closed discrete finite populations, the rearrangement of mass via aggregation and fragmentation events represents a random walk on a space of partitions [5,6,7,8,9,10,11]. If we adopt the view of partitions as a finite sample from a population of elements distributed by some extensive property (“mass”), it should come as no surprise that partitions appear in discrete stochastic processes in general, one example of which is a random walk on a discrete lattice [12].

The connection between partitions and thermodynamics may be summarized as follows. The set of partitions together with a probability measure form an ensemble. The distribution of elements in a partition is a random function whose probability is determined by the probability of the partition. In the thermodynamic limit, the most-probable distribution is overwhelmingly more probable than all others, and this behavior gives rise to thermodynamics. This connection has attracted the interest of the mathematical literature, which has focused on the definition of appropriate probability measures, the limit form of the distribution in the thermodynamic limit, and the mathematical conditions that ensure the existence of such a limit [13,14,15,16,17,18]. We take a different approach. We view the ensemble of partitions as a container of distributions—asymptotically of all distributions whose support is on the positive real axis—and the probability measure as a tool that can selectively extract any such distribution to deliver it as the most-probable distribution in the thermodynamic limit. In this sense, the space of partitions brings together two seemingly unrelated areas of mathematical physics, stochastic processes, and statistical thermodynamics. The point of contact between them is the probability distribution. To formalize this connection, we construct discrete finite ensembles, establish the equivalence between the microcanonical and the canonical ensemble, and study their combinatorial properties in the discrete finite domain.

The paper is organized as follows. In Section 2, we introduce the microcanonical table as a structured arrangement of partitions and define the probability measure. In Section 3, we define and study the discrete canonical ensemble and its partition function and establish equivalence with the microcanonical ensemble. In Section 4, we construct a random walk that samples the microcanonical table with the proper microcanonical probability and demonstrate with Monte Carlo simulations. In Section 5, we pass to the asymptotic limit and make full contact with thermodynamics. We discuss the results in Section 7 and, finally, summarize the conclusions in Section 8.

## 2. Microcanonical Ensemble of Partitions

An integer partition is a list of *N* integers whose sum is *M*. We consider ordered partitions (compositions, in the the language of number theory) and represent them by vector $c=(c_{1},c_{2}\cdots c_{N})$ with $c_{i}>0$ and $\sum_{i=1}^{N}c_{i}=M$. We construct the microcanonical ensemble $E_{M,N}$ as the set of all ordered partitions of integer *M* into *N* non-zero parts with *M* and *N* fixed. In our nomenclature, an order partition is a configuration, its elements are clusters, and their numerical value is the mass of the cluster. The number of clusters in the configuration is *N*, and their total mass is *M*. “Cluster” and “mass” are terms of the nomenclature, and they are not meant to assign physical properties to partitions, but rather serve as an analogy. Mass, in particular, stands for any additive property, for example mass, energy, volume, etc. The number of configurations in $E_{M,N}$ is [19]
(1)ΩM,N∗=M−1N−1.
A configuration is characterized by its distribution $n=(n_{1},n_{2}\cdots )$, where $n_{k}$ is the number of times integer *k* appears in configuration (number of clusters with mass *k*). All distributions in $E_{M,N}$ satisfy the conditions:(2)$$\begin{array}{c} \sum_{i}n_{i}=N,\;\sum_{i}in_{i}=M, \end{array}$$
which fix the zeroth- and first-order moments. The maximum possible cluster in the ensemble is M−N+1, but we let the upper limit in the summations go to *∞* with the understanding that $n_{i}=0$ for all i>M−N−1. The conditions in Equation (Equation 2) are necessary and sufficient: all distributions of configurations in $E_{M,N}$ satisfy these conditions, and conversely, all distributions that satisfy them represent configurations in $E_{M,N}$.

### 2.1. Microcanonical Table

A visual representation of the microcanonical ensemble is illustrated in Table 1 for M=7, N=5. We collect all ordered partitions of *M* into *N* parts to form a table (microcanonical table). The table has *N* columns and M−1N−1 rows, corresponding to the number of configurations of the ensemble. Every column contains the same list of numbers, k=1 through M−N+1, which may appear multiple times. We calculate this multiplicity as follows. The number of times element *k* appears in any one column is equal to the number of ways it may be combined with a list of N−1 positive integers that contain mass M−k to produce a configuration with *N* clusters and mass *M*. The set of these lists is the microcanonical ensemble $E_{M-k,N-1}$ and contains $\Omega_{M-k,N-1}^{*}$ ordered partitions. Therefore, the number of times integer *k* appears in each column of the microcanonical table is
(3)ΩM−k,N−1∗=M−k−1N−2,k=1…M−N+1.

The total number of elements in any column is $\Omega_{M,N}$. Therefore, we have the identity:(4)$$\begin{array}{c} \Omega_{M,N}^{*}=\sum_{k=1}^{M-N+1}\Omega_{M-k,N-1}^{*}. \end{array}$$
We also have
(5)$$\begin{array}{c} \Omega_{M,N}^{*}=\sum_{k=1}^{M-N+1}k\frac{\Omega_{M-k,N-1}^{*}}{\Omega_{M,N}^{*}}=\frac{M}{N}, \end{array}$$
which expresses the fact that the mean cluster in any column of the table is the same as in the entire table.

Returning to Table 1, the order of rows and columns is not important; permutations in their order produce the same list of configurations. The configurations in Table 1 are grouped by distribution, but in an otherwise arbitrary order. In this example, there are two distributions: $n_{A}=\left(4,0,1\right)$ with four monomers and one trimer, represented by five configurations; and $n_{B}=\left(3,2,0\right)$ with three monomers and one dimer, which is represented by ten configurations. In writing distributions in vector form, we understand that all omitted elements are zero.

### 2.2. Multiplicity

Configurations that are permutations of each other have the same distribution. The number of configurations with distribution n is equal to the multinomial factor of the distribution:(6)$$\begin{array}{c} n!=\frac{N!}{n_{1}!n_{2}!\cdots }. \end{array}$$
The multinomial factor represents the intrinsic multiplicity of the distribution, i.e., the number of configurations represented by the distribution. Suppose, however, that the elements of the configuration come in multiple internal variants, for example “color,” “shape,” “reactivity”, or a similar attribute. Variants increase the multiplicity of the distribution. If element *i* exists in $w_{i}$ variants and distribution n contains $n_{i}$ such elements, the multiplicity of the distribution increases by a factor $w_{i}^{n_{i}}$. When the variants of all elements are considered, the multiplicity of the distribution increases by the product of these factors:(7)$$\begin{array}{c} W\left(n\right)=\prod_{i=1}^{N}w_{i}^{n_{i}}. \end{array}$$
The total multiplicity is the product of W(n) with the intrinsic multiplicity n! and represents the statistical weight of the distribution in the ensemble:(8)$$\begin{array}{c} \mathrm{microcanonical}\;\mathrm{weight}\;\mathrm{of}\;\mathrm{distribution}=n!\;W\left(n\right)=N!\prod_{i}\frac{w_{i}^{n_{i}}}{n_{i}!}. \end{array}$$
W(n) is a functional of n that biases the multiplicity of the distribution in the ensemble. With W(n)=1, this multiplicity reduces to the intrinsic value n! and the partition function is reduced to $\Omega^{*}$ in Equation (Equation 1). By allowing *W* to vary between distributions, we effectively bias the weight of the distribution in the ensemble and gain a degree of freedom in manipulating the ensemble, which will prove useful.

It is possible to construct functionals that are more general than that in Equation (Equation 8), but the above form has several special properties that make it particularly useful. Among them is that *W* may be explicitly expressed in terms of the elements of the configuration. Given configuration $c=c_{1},c_{2}\cdots$ with distribution n, its multiplicity is
(9)$$\begin{array}{c} W\left(c\right)=\prod_{i}w\left(c_{i}\right)=W\left(n\right), \end{array}$$
where $w(c_{i})$ is the number of variants of size $c_{i}$.

### 2.3. Microcanonical Probability

The multiplicity of distribution is the number of times the distribution is represented in the ensemble in all ordered permutations of all variants of its elements. We define the microcanonical probability of the distribution as the ratio of its multiplicity over the total multiplicity in the ensemble:(10)$$\begin{array}{c} P^{\mu C}\left(n\right)=\frac{n!W(n)}{\Omega_{M,N}}=\frac{N!}{\Omega_{M,N}}\prod_{i}\frac{w_{i}^{n_{i}}}{n_{i}!}. \end{array}$$
The normalizing constant is the microcanonical partition function, equal to the sum of microcanonical weights in the ensemble:(11)$$\begin{array}{c} \Omega_{M,N}=\sum_{n}n!\;W\left(n\right). \end{array}$$
The probability of configuration c in the microcanonical table is
(12)$$\begin{array}{c} P^{\mu C}\left(c\right)=\frac{W(c)}{\Omega_{M,N}}=\frac{1}{\Omega_{M,N}}\prod_{i}w\left(c_{i}\right), \end{array}$$
and follows from Equation (Equation 10) and the fact that all n! permutations with the same distribution are equally probable.

To continue to use the microcanonical table as a visualization of the ensemble, we imagine each configuration in the table to represent W(n) actual configurations; this is the number of ways to build the configuration from all available variants, and it depends on the distribution of elements in the configuration. Therefore, every row of the table is understood to represent W(n) rows, where n is the distribution of the elements in that row.

### 2.4. Mean Distribution

One result that can be obtained very easily from the microcanonical table is the mean distribution of clusters, which we define as the frequency of element *k* in the ensemble. The mean frequency of clusters of size *k* in the ensemble is its frequency in any row of the microcanonical table. The number of times cluster size *k* appears in a row is equal to the number of times it can be combined with a complementary configuration of N−1 clusters and total mass M−k to form a configuration with *N* clusters and mass *M*:. The set of such complements forms the microcanonical ensemble $E_{M-k,N-1}$ and contains $\Omega_{M-k,N-1}$ elements. Since each cluster in the microcanonical table represents $w_{k}$ variants, the mean cluster distribution is
(13)$$\begin{array}{c} \frac{\langle n_{k}\rangle }{N}=w_{k}\frac{\Omega_{M-k,N-1}}{\Omega_{M,N}} \end{array}$$
for all M−N+1≥k≥1. A different derivation of the same result in the context of the zero-range process is given in [12].

## 3. Canonical Ensemble

We define the canonical configuration of length $N^{\prime }<N$ as the ordered subset of the first $N^{\prime }$ elements of a microcanonical configuration. The set of all canonical configurations of size $N^{\prime }$ forms the canonical ensemble $C_{N^{\prime }|M,N}$. The notation emphasizes the fact that the canonical ensemble is defined in the context of an enclosing microcanonical ensemble, and to distinguish between the two, we use primed variables for the canonical ensemble and unprimed for the enclosing microcanonical. The graphical construction of the canonical ensemble is illustrated in Table 2: Form the microcanonical table of the enclosing ensemble, and collect the first $N^{\prime }$ columns. This produces a canonical table with $N^{\prime }$ columns and $\Omega_{M,N}^{*}$ rows, each row representing a canonical configuration. In the illustration of Table 2, $N^{\prime }=2$. The remaining columns of the microcanonical table form the complementary canonical ensemble $C_{N-N^{\prime }|M,N}$. Since the ordering of columns in the microcanonical table is immaterial, we could pick any $N^{\prime }$ columns in any order, but for simplicity, we will continue to work with the first $N^{\prime }$ columns of the microcanonical table.

### 3.1. Canonical Probability

When a canonical configuration $c^{\prime }$ with distribution $n^{\prime }$ is cut from a microcanonical configuration c with distribution n, it leaves behind a complementary configuration $c^{\prime \prime }$ with distribution $n^{\prime \prime }$. The set of complements forms the complementary ensemble $C_{N-N^{\prime };M,N}$, and the two complements together form the enclosing microcanonical ensemble. The sum of two complementary distributions,
(14)$$\begin{array}{c} n=n^{\prime }+n^{\prime \prime }, \end{array}$$
is a member of the enclosing microcanonical ensemble $E_{M,N}$. In the language of thermodynamics, $C_{N-N^{\prime };M,N}$ is the system, its complement is the “bath”, and the enclosing microcanonical ensemble is the universe. We obtain the probability of canonical distribution by a combinatorial calculation. Distribution $n^{\prime }$ and its complement $n^{\prime \prime }$ form a microcanonical distribution n with microcanonical probability
(15)$$\begin{array}{c} P\left(n^{\prime }+n^{\prime \prime }\right)=\left(n^{\prime }+n^{\prime \prime }\right)!\frac{W\left(n^{\prime }\right)W\left(n^{\prime \prime }\right)}{\Omega_{M,N}}. \end{array}$$
There are $n^{\prime }!n^{\prime \prime }!$ ways out of $(n^{\prime }+n^{\prime \prime })!$ to combine $n^{\prime }$ and its complement; we obtain the canonical probability by summing the factor $P\left(n^{\prime }+n^{\prime \prime }\right)n^{\prime }!n^{\prime \prime }!/\left(n^{\prime }+n^{\prime \prime }\right)!$ over all complements $n^{\prime \prime }$. Using $P\left(n^{\prime }+n^{\prime \prime }\right)=\left(n^{\prime }+n^{\prime \prime }\right)!W\left(n^{\prime }+n^{\prime \prime }\right)/\Omega_{M,N}$ for the microcanonical probability of distribution $n^{\prime }+n^{\prime \prime }$, we obtain:(16)$$\begin{array}{c} P\left(n^{\prime }|N^{\prime };M,N\right)=\sum_{n^{\prime \prime }}^{}P\left(n^{\prime }+n^{\prime \prime }\right)\frac{n^{\prime }!n^{\prime \prime }!}{(n^{\prime }+n^{\prime \prime })!}=\frac{n^{\prime }!W\left(n^{\prime }\right)}{\Omega_{M,N}}\sum_{n^{\prime \prime }}^{}n^{\prime \prime }!W\left(n^{\prime \prime }\right). \end{array}$$
The summation on the far right is over the microcanonical multiplicities of all distributions with mass $M-M^{\prime }$ and number of particles $N-N^{\prime }$, and it is equal to the microcanonical partition function $\Omega_{M-M^{\prime },N-N^{\prime }}$. This leads to the following result for the canonical probability:(17)$$\begin{array}{c} P\left(n^{\prime }|N^{\prime };M,N\right)=n^{\prime }!W\left(n^{\prime }\right)\frac{\Omega_{M-M^{\prime },N-N^{\prime }}}{\Omega_{M,N}}. \end{array}$$
The canonical probability is proportional to the microcanonical weight of the distribution, but also depends on the partition functions of the complement and of the enclosing ensemble. To separate the effect of the enclosing ensemble, we write the canonical probability as
(18)$$\begin{array}{c} P\left(n^{\prime }|N^{\prime };M,N\right)=n^{\prime }W\left(n^{\prime }\right)\frac{\Omega_{M-M^{\prime },N-N^{\prime }}}{\Omega_{M,N-N^{\prime }}}\frac{\Omega_{M,N-N^{\prime }}}{\Omega_{M,N}}. \end{array}$$
Noting that $\Omega_{M,N-N^{\prime }}/\Omega_{M,N}$ is constant, we define the remaining portion of the expression on the right-hand side as the statistical weight of the canonical configuration:(19)$$\begin{array}{c} \mathrm{canonical}\;\mathrm{weight}\;\mathrm{of}\;\mathrm{distribution}=n^{\prime }!W\left(n^{\prime }\right)\frac{\Omega_{M-M^{\prime },N-N^{\prime }}}{\Omega_{M,N-N^{\prime }}}, \end{array}$$
We define the canonical partition function as the sum of canonical weights:(20)$$\begin{array}{c} Q_{N^{\prime };M,N}=\sum_{n^{\prime }}n^{\prime }!W\left(n^{\prime }\right)\frac{\Omega_{M-M^{\prime },N-N^{\prime }}}{\Omega_{M,N-N^{\prime }}}, \end{array}$$
with the summation over all canonical distributions. Applying the normalization condition on the canonical probability in Equation (Equation 18), we obtain
(21)$$\begin{array}{c} Q_{N^{\prime };M,N}=\frac{\Omega_{M,N}}{\Omega_{M,N-N^{\prime }}}. \end{array}$$
This establishes the relationship between the canonical and the microcanonical partition functions.

Unlike the microcanonical ensemble, in which the multiplicity and the statistical weight are equal, in the canonical ensemble, the two are not equal, but proportional to each other. The canonical multiplicity is the number of times a canonical distribution appears in the enclosing microcanonical table and clearly depends not only on the distribution itself, but also on the size of the enclosing ensemble: the same canonical distribution has higher multiplicity if it is removed from a larger microcanonical ensemble. By basing the definition of the canonical partition function on its statistical weight, rather than its multiplicity, we obtain a quantity that, in the asymptotic limit, is independent of the size of the enclosing ensemble. We derive the asymptotic limit in Section 5.

### 3.2. Mean Canonical Distribution

The mean distribution of the canonical ensemble is obtained trivially: it is the same as that of the canonical ensemble because all columns of the microcanonical table contain the same distribution of clusters:(22)$$\begin{array}{c} {\left.\frac{\langle n_{k}\rangle }{N}\right|}_{C}={\left.\frac{\langle n_{k}\rangle }{N}\right|}_{\mu C}=w_{k}\frac{\Omega_{M-k,N-1}}{\Omega_{M,N}}. \end{array}$$
The result is true for all $1\leq N^{\prime }\leq N\leq M$.

## 4. A Random Walk in the Microcanonical Space: The Exchange Reaction

### 4.1. Binary Exchange Reaction and Its Graph

We have defined a microcanonical space of configurations with an associated space of distributions. To experiment numerically with this space, we need a method to sample its elements with the correct probability. In this section, we construct such a method in the form of a random walk that visits configurations according to their microcanonical probability in Equation (Equation 10). We will then use the method in Section 6 to study the asymptotic behavior of the ensemble and the close relationship between the selection functional and the most-probable distribution.

We formulate the sampling process as a binary exchange reaction, a process that emulates reactions between physical clusters. Starting with a configuration *c*, we select two elements, *i* and *j*, and exchange the mass between them to create a new pair of clusters $i^{\prime }$ and $j^{\prime }$ under the mass-conserving condition $i+j=i^{\prime }+j^{\prime }$. The transfer produces a new configuration with the same number of particles and total mass, which we represent by the reaction:(23)$$\begin{array}{c} c\overset{\;\;i+j\to i^{\prime }+j^{\prime }\;\;}{\to }c^{\prime }. \end{array}$$
The binary exchange reaction establishes a network of transitions and adds a layer of connectivity between the elements of the microcanonical table. These connections form a graph whose nodes are configurations, and its edges represent individual exchange reactions. The graph has the following properties:1.It is bidirectional because the reverse of the exchange reaction is also a binary exchange reaction.2.It is connected: starting from any configuration, it is always possible to reach through a series of exchange reactions a configuration with one cluster of size M−N−1 (giant cluster) plus N−1 monomers; the mass of the giant cluster can then be distributed to the other clusters to produce any other configuration of the ensemble. Therefore, any configuration can be reached from any other.3.Every configuration is connected to (M−N)(N−1) other configurations. The maximum number of units that can be transferred from a cluster with mass *k* is k−1 (cluster masses cannot be zero). The total number of units that are available for exchange within a configuration is M−N, and since each cluster may transfer mass any of the other N−1 clusters, the number of connections that depart from any configuration is (M−N)(N−1).

Figure 1 shows the graph of binary exchange reactions in the microcanonical ensemble with M=7, N=5. The ensemble contains 15 configurations, each linked to 8 others via exchange reactions. The 15 configurations belong to two distributions, one with 4 monomers and 1 trimer, and one with 3 monomers and 2 dimers. A transition between two configurations corresponds to a transition between the two distributions; however, it is possible for the distribution to transition back to itself if the reaction produces a permutation of the initial configuration.

### 4.2. Random Walk on the Binary Exchange Reaction Graph

To sample the microcanonical table, we construct a random walk on the graph of binary exchange reactions, but in order to visit configurations with proper microcanonical probability, we must construct an appropriate transition probability. We begin by defining the equilibrium constant of the transition in Equation (Equation 23) as
(24)$$\begin{array}{c} K\left(c\to c^{\prime }\right)=\frac{W(c^{\prime })}{W(c)}=\frac{w_{i^{\prime }}w_{j^{\prime }}}{w_{i}w_{j}}, \end{array}$$
where $w_{i^{\prime }}$ and $w_{j^{\prime }}$ are the cluster weights of the products and $w_{i}$ and $w_{j}$ are those of the reactant clusters. According to Equation (Equation 9), the selection functional of the configuration is the product of the multiplicity of its elements, and since the product and reactant distributions differ only in the mass of elements that participate in in the reaction, the final result has the form of the familiar reaction equilibrium constant with the activities of chemical species replaced by the multiplicities of the elements of the configuration. We now set the transition probability for the reaction $c\to c^{\prime }$ by the Metropolis prescription:(25)$$\begin{array}{c} P\left(c\to c^{\prime }\right)=\left\{\begin{array}{cc} K(c\to c^{\prime }) & \mathrm{if}\;K(c\to c^{\prime })\leq 1\\ 1 & \mathrm{otherwise} \end{array}\right. \end{array}$$
The detailed balance condition:(26)$$\begin{array}{c} P\left(c\right)P\left(c\to c^{\prime }\right)=P\left(c^{\prime }\right)P\left(c^{\prime }\to c\right) \end{array}$$
is satisfied by the microcanonical probability in Equation (Equation 12). It follows [20] that the microcanonical probability is the stationary probability of the Markov process described by the transition probabilities from Equation (Equation 25): a random walk that starts from any configuration visits in the long run every configuration according to its microcanonical probability.

### 4.3. Monte Carlo Sampling

The exchange reaction can be simulated easily by the Monte Carlo method. We begin with an arbitrary configuration of *N* ordered integers, pick at random two elements, *i* and *j*, and replace them by two new numbers $i^{\prime }$ and $j^{\prime }$ such that $i^{\prime }+j^{\prime }=i+j$. To implement this numerically, we draw an integer random number 1<r<i+j and set $i^{\prime }=r$, $j^{\prime }=i+j-r$. If $i^{\prime }=i$, we reject the result and repeat with a new random number in order to avoid self transitions. However, this step is not necessary because self transitions do not alter the stationary distribution: the probability of self transition is 1/(z+1), where z=(M−N)(N−1) is the number of cross-transitions; this number is the same for all configurations and affects all distributions uniformly. This simplifies the simulation by allowing the transfer of any mass between two clusters.

With $w_{i}=1$, the selection functional is W(n)=1 for all n and distributions are sampled in proportion to their multinomial factor. By choosing the selection functional appropriately, we can bias the probability of the distribution towards any distribution of the ensemble. We demonstrate the effect of the selection functional for the case M=7, N=5. The microcanonical ensemble contains two distributions, $n_{A}=\left(4,0,1\right)$ and $n_{B}=\left(3,2,1\right)$ with $n_{A}!=5$, $n_{B}!=10$ and with corresponding probabilities 1/3 and 2/3, respectively. We construct the selection functional using $w_{i}=i^{\alpha }$. This leads to the following probabilities for the two distributions:(27)$$\begin{array}{c} P\left(n\right)=\frac{3^{\alpha }}{2^{2\alpha +1}+3^{\alpha }},\;P\left(n^{\prime }\right)=\frac{2^{2\alpha +1}}{2^{2\alpha +1}+3^{\alpha }}. \end{array}$$
Positive exponents favor the probability of $n_{A}$; in the limit α→∞, we obtain $P(n_{A})\to 1$, and similarly, with α→−∞, we obtained $P(n_{B})\to 1$. We illustrate this behavior in Figure 2 using Monte Carlo sampling to track the number of times each configuration is visited. With α=1, all configurations are visited with the same probability. With α>0, the configurations with distribution $n_{B}=\left(3,2,0\right)$ are visited more frequently than those with distribution $n_{A}=\left(4,1,1\right)$. With a<0, the bias shifts towards configurations with distribution $n_{A}$. In all cases, the configurations within the same distributions are equiprobable.

## 5. Asymptotic Limit

### 5.1. Microcanonical Thermodynamics

In the asymptotic limit M,N→∞ at fixed M/N, the discrete ensemble of distributions is quasi continuous in n and $\Omega_{M,N}$ may be treated as a continuous function of *M* and *N*. We define the parameters β and *q* as
(28)$$\begin{array}{c} \beta =\log\frac{\Omega_{M+1,N}}{\Omega_{M,N}}\to {\left(\frac{\partial \Omega_{M,N}}{\partial M}\right)}_{N},\;\log q=\frac{\Omega_{M,N+1}}{\Omega_{M,N}}\to {\left(\frac{\partial \Omega_{M,N}}{\partial N}\right)}_{M}. \end{array}$$
By Taylor’s expansion, we have
(29)$$\begin{array}{c} \log\Omega_{M-k,N-l}=\log\Omega_{M,N}-\beta k-l\log q, \end{array}$$
which is valid for k≪M, l≪N. Applying this result with l=1 to Equation (Equation 13), we obtain the mean distribution in the form:(30)$$\begin{array}{c} \frac{\langle n_{k}\rangle }{N}=w_{k}\frac{e^{-\beta i}}{q}. \end{array}$$
The factors β and logq are obtained from the conditions:(31)$$\begin{array}{c} \sum_{k}w_{k}\frac{e^{-\beta i}}{q}=1;\;\sum_{k}k\;w_{k}\frac{e^{-\beta i}}{q}=\frac{M}{N}, \end{array}$$
which express the fact that $\langle n_{k}\rangle /N$ is normalized to unity and its mean is M/N.

The mean distribution is uniquely determined by the weights $w_{k}$ and the mean cluster size. This further implies that β and logq, both of which are derivatives of $\log\Omega_{M,N}$, are functions of the ratio M/N, i.e., they are homogeneous in *M* and *N* with degree 0. It follows that $\log\Omega_{M,N}$ is homogeneous in *M* and *N* with degree one, then, by Euler’s theorem for homogeneous functions, we obtain
(32)$$\begin{array}{c} \log\Omega_{M,N}=M\beta +N\log q. \end{array}$$
A consequence of homogeneity is that the mean distribution in the asymptotic limit is overwhelmingly more probable than any other distributions. To see why, we calculate the log of the probability of distribution $\langle n\rangle$ by combining Equations (Equation 10) and (Equation 30):(33)$$\begin{array}{c} \log P\left(\langle n\rangle \right)=M\beta +N\log q-\log\Omega_{M,N}=0, \end{array}$$
whose right-hand side is zero by virtue of of Equation (Equation 32). Effectively, the selection functional picks out a single distribution from the microcanonical table and renders all others invisible. The inequality $\log P\left(n\right)<\log P\left(\langle n\rangle \right)$, which merely states that all other distributions of the ensemble are less probable that the mean distribution, can be expressed in the equivalent form:(34)$$\begin{array}{c} S\left(n\right)+\log W\left(n\right)\leq \Omega_{M,N}, \end{array}$$
where S(n) is the extensive Shannon functional of distribution n:(35)$$\begin{array}{c} S\left(n\right)=-N\sum_{i}\frac{n_{i}}{N}\log\frac{n_{i}}{N}. \end{array}$$
The inequality in Equation (Equation 34) is a statement of the second law: the functional on the left-hand side is maximized by the mean distribution in Equation (Equation 30). In the special case W(n)=1, Equation (Equation 34) states that the Shannon entropy of the mean distribution is the highest among all distributions in the ensemble.

### 5.2. Canonical Thermodynamics

We return to the canonical probability in Equation (Equation 17). Using Equation (Equation 32) to write $\log\Omega_{M-M^{\prime },N-N^{\prime }}=-\beta M^{\prime }-N\log q$, the canonical probability in the asymptotic limit is
(36)$$\begin{array}{c} P\left(n\right)=n^{\prime }!W\left(n^{\prime }\right)\frac{e^{-\beta M^{\prime }}}{q^{N^{\prime }}}. \end{array}$$
The canonical probability in the asymptotic limit is proportional to the microcanonical weight and the Boltzmann factor $e^{-\beta M^{\prime }}$, where $M^{\prime }$ is the total mass in the configuration. We obtained the canonical partition function by returning to Equation (Equation 21) in combination with Equation (Equation 32):(37)$$\begin{array}{c} Q_{N^{\prime },\beta }=q^{N^{\prime }}. \end{array}$$
Here, the notation $Q_{N^{\prime },\beta }$ implies that the canonical partition function does not depend on *M* and *N* individually, but on the intensive variable β, a function of the intensive ratio M/N. We now recognize the parameter *q*, which was defined as the partial derivative of $\Omega_{M,N}$, as the canonical partition function in a single column of the microcanonical table.

## 6. Construction of the Selection Functional

The microcanonical table asymptotically contains every normalized distribution $f_{k}$ with mean $\bar{k}=M/N$. We will construct a selection functional that picks any distribution $f_{k}^{*}$ from this space. We begin by setting
(38)$$\begin{array}{c} w_{k}^{*}=a\;e^{bk}f_{k}^{*}, \end{array}$$
where a>0 and *b* are arbitrary constants. It is a simple matter to confirm that this form satisfies Equation (Equation 30) with $\langle n_{k}\rangle /N=f_{k}^{*}$, q=a, and β=b. Since we are free to select *a* and *b*, we choose a=1, b=0, which gives $w_{k}^{*}=f_{k}^{*}$. Thus, we have a straightforward way to construct equilibrium constants for the exchange reaction so as to target any distribution as the equilibrium distribution. The only requirement is that the distribution has a finite mean.

We demonstrate the use of Equation (Equation 38) with two examples. In the first example, we consider the triangular distribution:(39)$$\begin{array}{c} f_{k}=0.0001\times \left\{\begin{array}{cc} 0 & \;\mathrm{if}\;x<100\;\mathrm{or}\;x>300\\ x-100 & \;\mathrm{if}\;100<x\leq \;200\\ 300-x & \;\mathrm{if}\;200<x\leq \;300 \end{array}\right. \end{array}$$
with support in 100≤k≤300 and mean $\bar{k}=200$. To simulate this distribution by the exchange reaction method, we use a list of N=1000 clusters with total mass M=200N so that the mean cluster in the list is 200, as in the distribution. We set $w_{k}=f_{k}$ if $f_{k}>0$ and $w_{k}=10^{-10}$ if $f_{k}=0$ (the cluster weight appears in the denominator of the equilibrium constant and cannot be zero). The simulation begins with the mass of all clusters set to $\bar{k}=200$. Figure 3a shows the results of the simulation after $4\times 10^{5}$ steps and demonstrates very good agreement with the distribution for which the selection functional was constructed.

For the second example, we construct a bimodal distribution formed as a mixture of two Gaussian distributions in equal proportions, one centered at $k_{1}=180$ with variance $\sigma_{1}^{2}=100$, the other one centered at $k_{2}=220$ with variance $\sigma_{2}^{2}=1000$. The mean of the bimodal distribution is $\bar{k}=200$. The simulation is again conducted with N=1000 particles with total mass M=200N, starting with all particles at mass $\bar{k}=200$. Figure 3b shows that, in this case, as well, the simulated distribution converges to the distribution for which the $w_{k}$ values were designed.

## 7. Discussion

The set of integer partitions illustrates the structure of thermodynamic ensembles and the emergence of the thermodynamic limit. The table of ordered partitions represents the microcanonical ensemble; any number of columns extracted from this table forms a canonical ensemble. The equivalence between these ensembles is established by the elementary property that all columns of the table of ordered partitions contain the same list of clusters. The classical proof requires the study of fluctuations in the thermodynamic limit. Here, we have established equivalence in the discrete finite domain.

We have assigned probabilities in proportion to the microcanonical functional n!W(n). The standard mathematical treatment [14] does not distinguish between n! and W(n) individually. This is not a trivial mathematical detail. The multinomial coefficient arises because we treat permutations in the order of the partitions as distinct from each other. It is only when we take the order of partitions into consideration that the microcanonical table presented a structure with well-defined combinatorial properties. The reference ensemble is defined by the condition W=1. This renders all ordered partitions equally probable; the multiplicity of the distribution is given by the multinomial coefficient, and the most-probable distribution is exponential. This is Boltzmann’s derivation of combinatorial entropy [21] (p. 55). Thus, we make contact with a fundamental result of statistical mechanics. The uniform selection functional is the mathematical statement of the postulate of equal a priori probabilities. The most-probable distribution in this case is the distribution with the maximum multinomial coefficient, and since its logarithm is the entropy functional, we concluded that the most-probable distribution in the microcanonical table under a uniform prior is the maximum entropy distribution. The selection functional biases the probability of configurations relative to that in the reference ensemble and can be designed to deliver any distribution present in the ensemble as the most-probable distribution in the thermodynamic limit. This was shown previously within an abstract space of distributions [22]; we have obtained the same result in the space of distributions contained in the microcanonical table. The exchange reaction serves to highlight the connection between integer partitions and thermodynamics in a physical way. We imagine integer partitions to represent a collection of clusters and the exchange reaction to represent a reversible reaction between them. The selection functional is the chemical “activity” of the clusters and determines the equilibrium constant of the reaction. Importantly, only the selection functional appears in the equilibrium constant. This is yet another reason for treating the multinomial coefficient and the selection functional as two separate functionals: The multinomial factor accounts for the random selection of the clusters chosen to react and represents what in thermodynamics we call an “ideal” system; the selection functional expresses deviations from ideality by imposing a bias relative to a purely random system. This is most clearly expressed in Equation (Equation 30), in which the factor $w_{k}$ can be viewed as a correction to the exponential distribution $e^{-\beta k}/q$ because of nonidealities.

In a further departure from the standard mathematical literature, we have provided an exact treatment of discrete finite ensembles of partitions, including ensembles that are too small to be treated as continuous. The microcanonical probability is given in Equation (Equation 10) and the canonical probability in Equation (Equation 17). The mean distribution is given by Equation (Equation 22), while the most-probable distribution is defined by the condition $\max_{n}n!W(n)$ and satisfies the second law in Equation (Equation 34) as a strict inequality. All of these results apply to ensembles of any size. The thermodynamic limit requires $n_{i}$ to be large enough (equivalently, M,N≫1) that it may be treated as a continuous variable. To demonstrate the transition from the discrete/finite to discrete/infinite and, finally, to continuous/infinite domain, we consider the case W(n)=1 for which the partition function is given in Equation (Equation 1). Applying Equation (Equation 13) with $w_{i}=1$, we obtain the mean distribution in the form:(40)nkN=M−k−1N−1M−1N−1→1x¯−1x¯x¯−1k→e−k/x¯x¯,
with $\bar{x}=M/N$. The first result applies to all *M*, *N*, M−N+1≥k≥1; the second result applies to M>N≫1 (*M*); the last result is true for M≫N≫1. Under the last condition, the cluster mass may be treated as a continuous variable, and thus, we recover the exponential distribution as the most-probable distribution in the unbiased ensemble.

## 8. Conclusions

In summary, we have formulated the combinatorial properties of the ensemble of integer partitions. By considering ordered partitions, we obtain an organized tabulation such that the table itself is a microcanonical ensemble and its rows are canonical ensembles. The ensemble is a container of distributions and contains a discrete finite sample of all distributions with finite mean and support on the real axis. The natural multiplicity of the distribution in the ensemble is its multinomial coefficient, and its logarithm is the entropy of the distribution. By further biasing the probability of the distribution via the selection functional, we can select any distribution from this ensemble. In the asymptotic limit, the space of distributions obeys thermodynamics: it gives rise to a distribution that is overwhelmingly more probable than all others and whose parameters satisfy the familiar thermodynamic relationships. We may view the ensemble of partitions as a generic template for stochastic processes. The central quantity in any stochastic process is the probability distribution of a stochastic variable, which may be thought to arise from the ensemble of partitions under a suitable functional. The determination of this functional is the challenge in this approach, but if this functional could be identified, we would obtain a rigorous thermodynamic treatment of the process. A few examples have been given in the literature of population balances [5,10,11,23]. We suggest that this approach can be extended beyond population balances to stochastic processes in general.

## Figures and Tables

**Figure 1 entropy-25-00385-f001:**
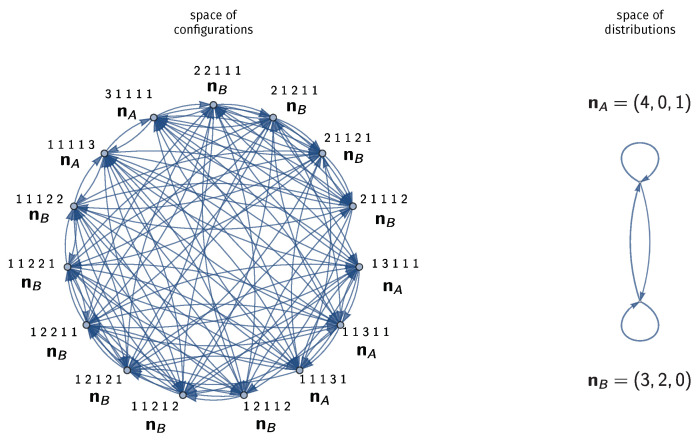
(**Left**) Configurations undergoing an exchange reaction represent transitions that visit the space of configurations uniformly. A random walk on this space visits each configuration the same number of times. (**Right**) The corresponding transitions between distributions visit each distribution n in proportion to its multinomial factor n!.

**Figure 2 entropy-25-00385-f002:**
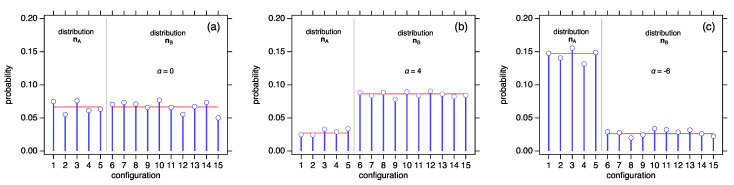
Exchange reactions in the ensemble M=7, N=5, visit the 15 configurations of the ensemble in proportion to the cluster weight $w_{k}=k^{\alpha }$. Configurations 1–5 represent distribution $n_{A}=\left(4,1,1\right)$; Configurations 6–15 represent distribution $n_{A}=\left(3,2,0\right)$ (configurations are numbered in the order they appear in Table 1). (**a**) α=0: all configurations are visited with equal probability; (**b**) α=4: configurations of distribution $n_{A}$ are visited more frequently; (**c**) configurations of distribution $n_{B}$ are visited more frequently. Lines show the theoretical probability calculated as P(n)/n!, where n is the distribution of the clusters in the configuration.

**Figure 3 entropy-25-00385-f003:**
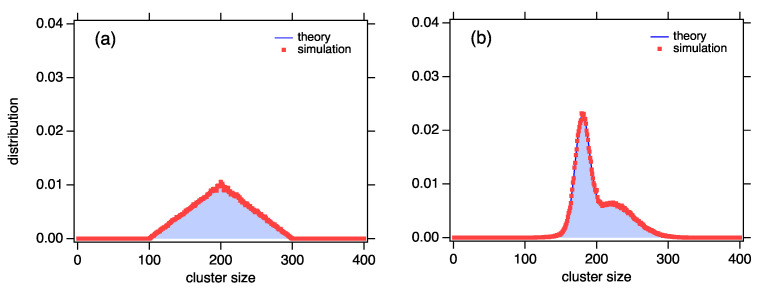
(**a**) Triangular distribution; (**b**) bimodal distribution of two Gaussian distributions. Symbols are MC simulations of the exchange reaction with $w_{k}=f_{k}$, where $f_{k}$ is the triangular or the bimodal distribution. In both cases, the simulation agrees very well with the corresponding distribution.

**Table 1 entropy-25-00385-t001:** Microcanonical ensemble with M=7, N=5. (**a**) Microcanonical table of configurations ($m_{i}$ is the *i*th element of the configuration); (**b**) list of distributions in the ensemble ($n_{i}$ is the number of *i*-mers in the configuration). The table of distributions is a more concise representation of the ensemble of configurations with each distribution representing n!W(n) distinct configurations.

**(a) Configurations**				
** *m* ${}_{1}$ **	** *m* ** ${}_{2}$	** *m* ${}_{3}$ **	** *m* ${}_{4}$ **	** *m* ${}_{5}$ **
3	1	1	1	1
1	3	1	1	1
1	1	3	1	1
1	1	1	3	1
1	1	1	1	3
2	2	1	1	1
2	1	2	1	1
2	1	1	2	1
2	1	1	1	2
1	2	2	1	1
1	2	1	2	1
1	2	1	1	2
1	1	2	2	1
1	1	2	1	2
1	1	1	2	2
**(b) Distributions**				
** *n* ${}_{1}$ **	** *n* ${}_{2}$ **	** *n* ${}_{3}$ **	** *n* ${}_{4}$ **	** n! **
4	0	1	0	5
3	2	0	0	10

**Table 2 entropy-25-00385-t002:** Canonical partitioning of microcanonical ensemble $E_{M,N}$ with M=7, N=5. The shaded configurations form canonical ensemble $C_{N-N^{\prime }|M,C}$ with $N^{\prime }=2$. The unshaded portion constitutes the complementary ensemble $C_{N-n^{\prime }|M,N}$.

Canonical Configurations				
$m_{1}^{\prime }$	$m_{2}^{\prime }$	$m_{1}^{\prime \prime }$	$m_{2}^{\prime \prime }$	$m_{3}^{\prime \prime }$
1	1	3	1	1
1	1	1	3	1
1	1	1	1	3
1	1	2	2	1
1	1	2	1	2
1	1	1	2	2
2	1	2	1	1
2	1	1	2	1
2	1	1	1	2
1	2	2	1	1
1	2	1	2	1
1	2	1	1	2
2	2	1	1	1
3	1	1	1	1
1	3	1	1	1

## Data Availability

Not applicable.

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
