# Peer review of "Combinatorics and Statistical Mechanics of Integer Partitions"

_entropy, 2023, doi:10.3390/e25020385_

Round 1

Reviewer 1 Report

In the present manuscript the author reviews the problem of the statistical mechanics and thermodynamic properties of the set of integer partitions, focusing on the case where partitions are ordered. The formalism for both microcanonical and canonical ensembles is derived, and the main thermodynamic relations (entropy, Gibbs equation, ...) are provided.

I find the topic addressed is attractive and could fit quite well the interests of Entropy readers. However, the manuscript in its present version suffers from several flaws in its style and presentation that should be amended before a decision can be taken. In particular (in order of importance):

ia) In its present form, the work is merely presented as a mathematical exercise. I think that the manuscript would benefit much from a discussion (in the Introduction section) about potential examples/applications that the case of ordered integer partitions may cover, and this could also attract the interest of a wider audience.

ib) Also, the introduction (as well as the discussion) section should state more clearly what are the main novelties/contributions that the present work offers in comparison to previous ones where the ensemble properties of integer partitions have been considered. If the only difference is on the ordering of partitions (as the author seem to suggest in the manuscript) then a more thorough discussion about this case should be provided in order to justify its physical interest.

ii) Regarding the structure of the manuscript, I think that the motivation and interest of Section IV is not clearly stated in the text. As I understand, its interest is justified later in Section VI when some examples of selection functionals are worked out. So that, to preserve the line of reasoning in the text I would suggest the author to move Section IV to an appendix, or present it as an auxiliary Section (Numerical Methods, or similar) of Section VI.

iii) Regarding the numerical results, there is a very strange point in the center of the distribution in Figure 3a that is completely out of the expected distribution. It looks as if there is an artefact due to the method used to discretize the distribution coming from the simulations. This should be corrected or discussed.

iv) The writing of the manuscript should be revised as it contains several typos. Here is a list of those I have spotted:
-S1, Line 88: We introduces -> We introduce
-S1, Line 89: In Section 3 We... -> In Section 3 we...
-1st paragraph of S2: the configuration in... -> the configuration is...
-S2, Table 1 and line 78: n_2 for case B should be equal to 2, not 1
-1st paragraph of S4.2: but i order... -> but in order...
-S7, line 224: about the the relationship... -> about the relationship...

Author Response

ia) In its present form, the work is merely presented as a mathematical exercise. I think that the manuscript would benefit much from a discussion (in the Introduction section) about potential examples/applications that the case of ordered integer partitions may cover, and this could also attract the interest of a wider audience.

ANSWER: We have revised the introduction and added a paragraph to clearly motivate the study and to connect it to other problems of interest. 

ib) Also, the introduction (as well as the discussion) section should state more clearly what are the main novelties/contributions that the present work offers in comparison to previous ones where the ensemble properties of integer partitions have been considered. If the only difference is on the ordering of partitions (as the author seem to suggest in the manuscript) then a more thorough discussion about this case should be provided in order to justify its physical interest.

ANSWER: There are several points of departure, and these are now highlighted inthe newly added section Discussion. 

ii) Regarding the structure of the manuscript, I think that the motivation and interest of Section IV is not clearly stated in the text. As I understand, its interest is justified later in Section VI when some examples of selection functionals are worked out. So that, to preserve the line of reasoning in the text I would suggest the author to move Section IV to an appendix, or present it as an auxiliary Section (Numerical Methods, or similar) of Section VI.

ANSWER: We believe that Section 4 is important on its own merit. In Section 2 we constructed a probability space on the microcanonical table. In Section 4 we construct a network of connections that allow us to conduct a random walk that visits elements of that space with proper probabilities. We use this network in Section 5 to demonstrate by simulation that the microcanonical table contains every normalizable distribution. The random walk (exchange reaction) imposes a dynamic process on top of the static ensemble. We believe it is important to *prove* that the  probability of the dynamic space converges to that of the static space. Through this association we recognize $W$ as the "thermodynamic activity" of the configuration. It is precisely through this connection that we can understand the emergence of thermodynamics in the asymptotic limit, even though the elements of the ensemble are ordered partitions, not physical particles. 

iii) Regarding the numerical results, there is a very strange point in the center of the distribution in Figure 3a that is completely out of the expected distribution. It looks as if there is an artefact due to the method used to discretize the distribution coming from the simulations. This should be corrected or discussed.

ANSWER: The reviewer is very perceptive -- indeed this is an artifact. The initial distribution is a delta function at mass 200. The artifact represents leftover clusters in this mass that have not equilibrated yet. We have run longer simulations until this peak is fully absorbed by the equilibrium distribution. 

iv) The writing of the manuscript should be revised as it contains several typos. Here is a list of those I have spotted:

ANSWER: We thank the reviewer for catching these typos. We have corrected them. 

Reviewer 2 Report

The paper is an interesting presentation of combinatorial
questions about the way of decomposing an integer M into a sum of
N positive integers. However the connection with thermodynamics,
not really discussed, does not add, and certainly does not
stress, novelties with respect to the classic discussion started
by Boltzmann. Therefore I cannot recommend its publication as a
research work.

Author Response

The paper is an interesting presentation of combinatorial
questions about the way of decomposing an integer M into a sum of
N positive integers. However the connection with thermodynamics,
not really discussed, does not add, and certainly does not
stress, novelties with respect to the classic discussion started
by Boltzmann. Therefore I cannot recommend its publication as a
research work.

ANSWER: The novelty with respect to the classic discussion started
by Boltzmann is that the entire network of thermodynamic relationship, including the inequality of the second law, appear naturally in the mathematical space of ordered partitions -- no molecules required. The mathematical object that produces this behavior is the selection functional, a functional that biases the sampling of configuations. We have added language to that effect. 

Reviewer 3 Report

I found to this be an interesting analysis and, overall, well-presented. Essentially this is a discussion of the statistical mechanics of discrete systems, which could be of potential interest in a wide range of areas (e.g., small systems, clusters, etc.), some of which the author is already aware. I recommend publication essentially as is, though the author should in one case, and may in the others, consider the following:

1. If I am not mistaken, Table 1b, along with the discussion about it at the bottom of page 3, has an error. The distribution should be 3,2,0, or three monomers and two dimers, which leads to the multiplicity of 10 using eq (6).

2. The use of the integer partitions, along with eq (2), reminds me very much of the problem of counting occupation numbers in quantum statistics. Does the author think that this work, or some aspect of it discussed here, has some bearing on the standard counting of microstates for Bose-Einstein statistics (where more than one particle can occupy a given state) and for Fermi-Dirac statistics (where no more than one particle can occupy a state)?

3. Perhaps my "non-surprise" that this analysis obeys thermodynamics in the asymptotic limit is misplaced. I expect that any distribution that becomes sharply peaked about an average value, with an exceedingly low relative fluctuation about the mean or most probable state, will provide consistency with macro thermodynamics. And so, is such a result that is found here so unexpected?

4. I liked the introduction of the canonical ensemble through it being a "subset" of a larger microcanonical ensemble. A large enough system yielded the usual Boltzmann weighting for this canonical ensemble. But are there any interesting results or insights that follow here when the system (or particularly the surrounding "bath") is not in the large system limit? Specifically, how does the "temperature" of the system and bath behave when they are both small?

Author Response

1. If I am not mistaken, Table 1b, along with the discussion about it at the bottom of page 3, has an error. The distribution should be 3,2,0, or three monomers and two dimers, which leads to the multiplicity of 10 using eq (6).

ANSWER: The reviewer is correct -- we corrected the typo

2. The use of the integer partitions, along with eq (2), reminds me very much of the problem of counting occupation numbers in quantum statistics. Does the author think that this work, or some aspect of it discussed here, has some bearing on the standard counting of microstates for Bose-Einstein statistics (where more than one particle can occupy a given state) and for Fermi-Dirac statistics (where no more than one particle can occupy a state)?

ANSWER: Indeed all of these probability distributions can be obtained with an appropriate choice of the selection functional. Some in fact have been discussed in Ref 14 & 15. Our goal was more general, to show that *any* probability distribution can be chosen to arise as the most probable distribution in the thermodynamic limit. It is for this reason that we view the ensemble of partitions as a template for stochastic processes in general. 

3. Perhaps my "non-surprise" that this analysis obeys thermodynamics in the asymptotic limit is misplaced. I expect that any distribution that becomes sharply peaked about an average value, with an exceedingly low relative fluctuation about the mean or most probable state, will provide consistency with macro thermodynamics. And so, is such a result that is found here so unexpected?

ANSWER: One can indeed argue that whenever a distribution becomes exceedingly sharp about its average, thermodynamics should be expected to arise. But this expectation alone is not very useful: in the central limit theorem we are dealing with a sharply peaked Gaussian, but this does not tell us how to apply thermodynamics to CLT, what the microstate is, or how to calculate its probability and the partition function. We show exactly how to do this in the case of the integer partitions. 

4. I liked the introduction of the canonical ensemble through it being a "subset" of a larger microcanonical ensemble. A large enough system yielded the usual Boltzmann weighting for this canonical ensemble. But are there any interesting results or insights that follow here when the system (or particularly the surrounding "bath") is not in the large system limit? Specifically, how does the "temperature" of the system and bath behave when they are both small?

ANSWER: An interesting point, we have added a paragraph in the Discussion of the revised manuscript to show how the size of the system affects its thermodynamics. 

Round 2

Reviewer 1 Report

After having read the new version of the manuscript, my impression is that the authors have made a reasonable to effort to address the concerns and criticisms raised in my previous report.

There is, however, still one small point that may be reformuIated in the new version. Regarding my recommendation to move Section 4 to an Appendix, the authors argue that this section is important within the line of reasoning in the manuscript. I am willing to accept their argument, but still my opinion is that the justification for including this Section is only made clear later in Sections V and VI. So that, to facilitate reading of the manuscript, I urge the authors at least to advance at the beginning of Section IV part of the analysis that will be carried out later, and/or to illustrate more clearly the practical interest and the intention that Section IV has.

Once this point has been properly implemented I think the current manuscript could be accepted for publication.

Author Response

After having read the new version of the manuscript, my impression is that the authors have made a reasonable to effort to address the concerns and criticisms raised in my previous report.

There is, however, still one small point that may be reformuIated in the new version. Regarding my recommendation to move Section 4 to an Appendix, the authors argue that this section is important within the line of reasoning in the manuscript. I am willing to accept their argument, but still my opinion is that the justification for including this Section is only made clear later in Sections V and VI. So that, to facilitate reading of the manuscript, I urge the authors at least to advance at the beginning of Section IV part of the analysis that will be carried out later, and/or to illustrate more clearly the practical interest and the intention that Section IV has.

Once this point has been properly implemented I think the current manuscript could be accepted for publication.

>> I appreciate the reviewer's concern and after rereading the manuscript I agree.  In the most recent revision, Section 4 begins with the following introductory paragraph that motivates the development of the random walk and makes reference to subsequent section 6 where this walk is used:

    We have defined a microcanonical space of configurations with an associated space of distributions. To experiment numerically with this space we need a method to sample its elements with the correct probability. In this section we construct such a method in the form of a random walk that visits configurations according to their microcanonical probability in Eq. 10. We will then use the method in Sections 5 and 6 to study the asymptotic behavior of the ensemble and the close relationship between the selection functional and the most probable distribution. 

Reviewer 2 Report

There seems to be no real improvement with respect to the first version

Author Response

There seems to be no real improvement with respect to the first version

I thank the reviewer.